# Cold Ironing: Socio-Economic Analysis in the Port of Genoa

**Monica Canepa** [1,*] , **Fabio Ballini** [1] , **Dimitrios Dalaklis** [1] , **Giampaolo Frugone** [2] **and Davide Sciutto** [3]

1   Maritime Energy Management Specialization, World Maritime University, 20124 Malmö, Sweden
2   Independent Researcher, 16100 Genoa, Italy
3   Western Ligurian Sea Port Authority, 16124 Genoa, Italy
*   Correspondence: moc@wmu.se

**Abstract:** *Background:* The emissions of ships in ports are increasingly becoming an issue, and particularly those of $NO_x$, $SO_x$, and PM, rather than $CO_2$. One realistic solution for reducing GHG emissions is cold ironing technology. This paper highlights the socio-economic impact of increasing utilization of cold ironing in the port of Genoa, quantifying the achievable emissions reductions and estimating the effects over a defined time horizon. *Methods:* This research employs an interdisciplinary approach and uses quantitative data with a bottom-up approach for data collection. For the quantification of health costs, reference was made to the CAFE program, which defines a calculation method to estimate the external costs generated by polluting agents such as $NH_3$, $NOx$, $SO_2$, VOCs, and PM2.5. *Results:* Analysis of results shows the significant importance of relying on cold ironing and the importance of renewable port generation. Health cost savings using cold ironing and a different mix of generations are calculated, and these results strongly signal the importance of cold ironing. *Conclusions:* Cold ironing is indeed an effective anti-pollution measure. Its use to reduce polluting emissions is to be strongly recommended. Investments are cost-effective versus health costs and are sustainable by all parties.

**Keywords:** cold ironing; GHG emissions; social costs

## 1. Introduction

In recent decades, the environmental question has become progressively more and more critical from different points of view. The most important one is the awareness of the extensive adverse effects caused by pollution.

This work addresses the specific case of the port of Genoa; the effects of pollution are analyzed using data collected in different scenarios. These data are used for the analysis of the consequences of the pollution.

Global warming is a challenge for the Earth, and maritime transport plays a critical role in greenhouse gas (GHG) emissions [1].

At the international level, environmental protection has been a priority for a long time; an example of this priority is the Kyoto Protocol of 1997, which aimed at reducing greenhouse gases.

Concerning maritime transport, it has been estimated by many researchers that 70% of associated global emissions occur within a radius of 400 km from ports; therefore, the negative environmental impact on coastal areas is evident [2]. Over the next decade, emissions of the sector are expected to increase by 4% per year, so transport is considered among the primary sources of atmospheric pollution in Europe. Therefore, it is reasonable that local air pollution is one of the main concerns of port authorities [3].

The growth of the maritime sector has been the object of growing attention to the effects of pollutants released into the environment. The fourth GHG (greenhouse gas) report of the IMO in 2020 [4] includes carbon intensity estimates for the first time, indicating that the average carbon intensity across international shipping was 20–30% better in 2018 than in the Initial Strategy's baseline year of 2008. According to the study, and based on several

long-term economic and energy scenarios (excluding the long-term effects of COVID-19), shipping emissions are predicted to rise from about 90% of 2008 levels in 2018 to 90–130% of 2008 levels by 2050, without any additional measures.

The main consequences of ship emissions are acidification and eutrophication of the environment, resulting in the formation of poisonous compounds that cause lung infiltration, blood poisoning, heart failure, and, therefore, premature deaths [5].

In light of these problems, the International Maritime Organization (IMO) has established a worldwide limit on sulfur emissions in non-SECA areas from 3.5% to 0.5% (starting date 1 January 2020) under rule 14.1.3 of Annex VI of the MARPOL convention, in conjunction with other documents such as the first IMO strategy to reduce greenhouse gas emissions adopted in April 2018 (IMO, 2018).

Appendices to the Convention SECA, which covers about 0.3% of world waters [6], include and refer to areas in the Baltic Sea, the North Sea, North America, and the Caribbean of the United States (IMO, 2018a). In the SECA areas, more restrictive limits have been set. A sulfur content of 0.1% was set as acceptable from 1 January 2015. It is claimed that implementing the SECA regulation in the Baltic Sea [7] every year "has saved 500–1000 premature deaths annually, 500 non-fatal myocardial infarctions and 500 cases of stroke".

The European Union set the goal of reducing its greenhouse gas emissions by at least 20% by 2020, and increasing the share of the utilization of renewable energy by at least 20% to achieve energy savings of 20% or more. The 2020 Energy Strategy (European Commission, 2010) has been implemented to achieve these objectives. In addition, measures have been adopted at the European level to combat maritime pollution.

A directive of great importance for this purpose is 1999/32/CE, which aims to reduce the sulfur content of certain liquid fuels. It has been substantially amended several times; the last amendment was through Directive 2012/33/EU of 21 November 2012 on the content of sulfur of fuels for marine use, which entered into force on 17 December 2012.

An EC report shows that outcomes from the implementation of Directive 1999/32/EC since the last revision of 2012 (now codified as Directive (EU) 2016/802) are positive because there has been a significant reduction in $SO_2$ concentrations in coastal regions. This reduction is true in the SECAs, while keeping the overall economic effects to a minimum (EC Report 2018).

Regulation (EU) 2016/1628 is in force, and concerns measures to be taken against the emission of gaseous and particulate pollutants produced by internal combustion engines installed on non-road mobile machines.

Directive (EU) 2016/802 of the European Parliament and of the Council of 11 May 2016 addresses marine fuel consumption, sulfur content, implementation of cold ironing in European ports, and setting of SECAs in the North and Baltic Seas, with a maximum of 0.1% sulfur content in the marine fuel consumed by vessels in these areas and areas outside SECAs (3.50%). In addition, since 1 January 2010, the maximum sulfur content in fuels must not exceed 0.1% for ships docked in all European ports.

Cold ironing (i.e., the process of generation and provision of shoreside electrical power to a ship while at berth) is generally regarded as one of the key factors in reducing pollution originating from ships berthed in ports. This technology started developing around ten years ago, especially in North America and Europe, and it is now spreading worldwide.

The CAFE program was established to support the European Commission's development of the Thematic Strategy on air pollution, the Directive on Ambient Air Quality and Cleaner Air for Europe, and its Impact Assessment. CAFE parameters have been adopted in this paper to perform economic analysis. A review of the direct costs generated by transport is carried out, with the unique perspective of naval transport. Particular attention has been given to the quantitative description of health costs. Figure 1 shows the number of ports with OPS facilities in December 2020 in the European Economic Area.

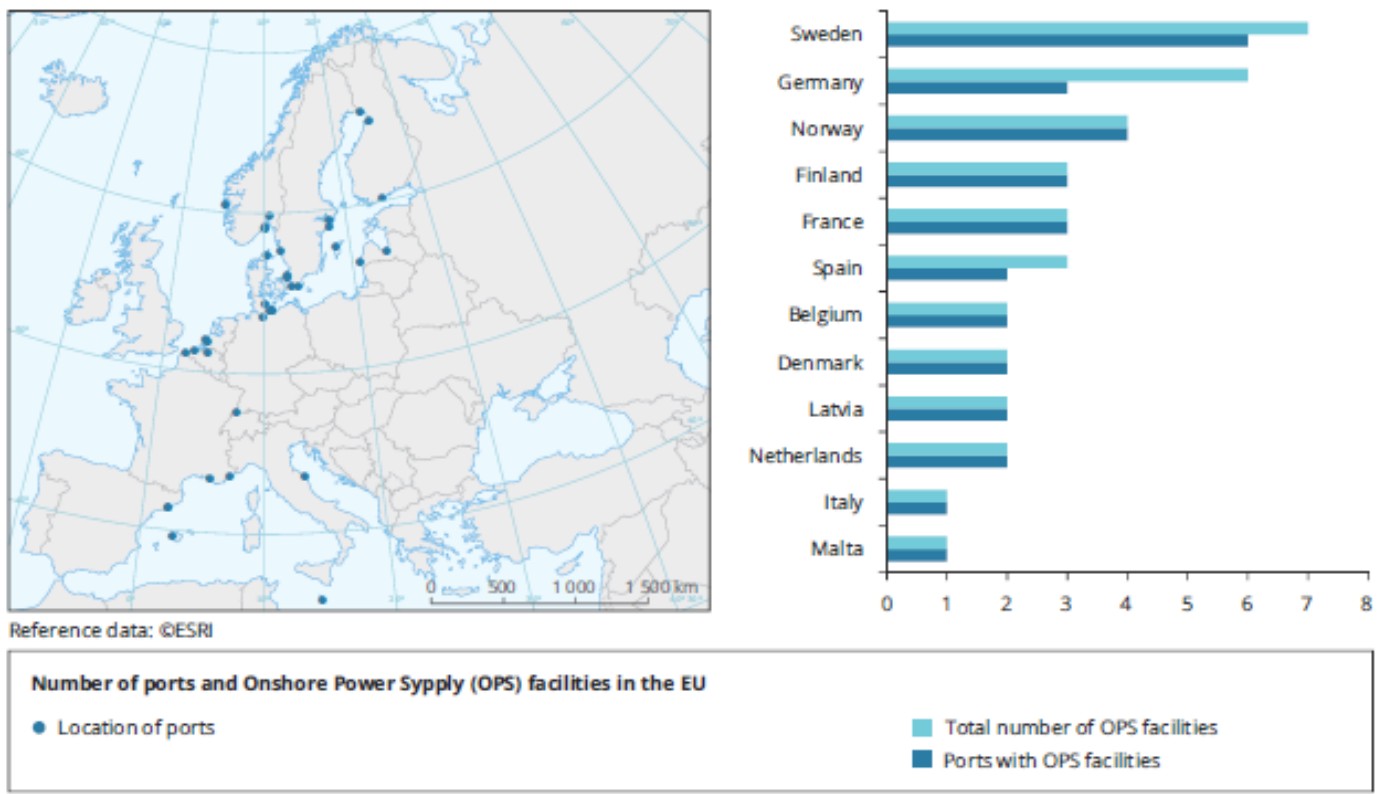

**Figure 1.** Number of ports and high-voltage OPS facilities in the European Economic Area (as of December 2020). Source: EAFO (2020) (from EMTER Report 2021).

Referring to 2021, it is noted that more than 31 EU ports and about 10% of vessels calling at EU ports have the capability to use OPSs.

## 2. Objectives

The assessment of the socio-economic impact of cold ironing in the port of Genoa and in the VTE (Voltri Terminal Europe) quantifies the emissions reduction achievable, and consequently estimates, in light of necessary investments, the impacts over a defined time horizon. The achievement of these objectives requires the analysis and solution of the following research questions.

- What socio-economic impact has derived from the emissions of ships that visited the VTE in 2022?
- What are the external costs generated by transport? In particular, what are the estimated health costs due to pollutant emissions?
- What socio-economic impact derives from adopting cold ironing in the assumed scenarios?
- How much are the benefits and social costs?

## 3. Reference Legislation and Regulations

In recent decades, the environmental question has become progressively more critical from different points of view due to the awareness of the adverse effects caused by pollution. At the international level, environmental protection has become a priority; an example is the use of legal instruments, such as the Kyoto Protocol of 1997, aimed at reducing greenhouse gases. There was a willingness to adapt to the requirements of the aforementioned protocol from the Community and national points of view.

With reference to maritime transport, it is estimated that 70% of global emissions occur within a radius of 400 km from the coast. Therefore, the negative environmental impact to the detriment of coastal areas is evident. Over the next decade, the sector's emissions are

expected to increase to 4% per year. This type of transport is accounted for as the primary source of atmospheric pollution in Europe.

Since the 20th century, the growth of the maritime sector has been accompanied by growing attention to the effects of pollutants released into the environment. The fourth GHG of the IMO in 2020 indicated that, from 2012 to 2018, the greenhouse gas (GHG) emissions from all types of shipping (international, domestic, and fishing) have increased by 9.6%, rising from 977 million tons to 1076 million tons, and include carbon dioxide ($CO_2$), methane ($CH_4$), and nitrous oxide ($N_2O$). In 2012, $CO_2$ emissions accounted for 962 million tons, while in 2018, this figure grew by 9.3% to 1056 million tons. The shipping sector's share of global anthropogenic emissions has also risen from 2.76% in 2012 to 2.89% in 2018.

As previously noted at the beginning of the paper, the main consequences deriving from ship emissions are acidification and eutrophication of the environment, which result in the formation of poisonous compounds. These cause lung infiltration, blood poisoning, heart failure, and, therefore, premature deaths [8–10].

In light of these problems, the International Maritime Organization (IMO) has established a worldwide limit on sulfur emissions in non-SECA areas from 3.5% to 0.5% (starting date 1 January 2020) under rule 14.1.3 of Annex VI of the MARPOL convention and other software tools, such as the first IMO strategy to reduce greenhouse gas emissions adopted in April 2018 (IMO, 2018).

Appendices to the Convention on Sulfur Emission Control Areas (SECA) relate to about 0.3% of world waters [11], including areas in the Baltic Sea, the North Sea, North America, and the Caribbean of the United States (IMO, 2018a). In the SECA areas, more restrictive limits provide an acceptable sulfur content of 0.1% from 1 January 2015. It is claimed that implementing the SECA regulation in the Baltic Sea [7] every year "has saved 500–1000 premature deaths annually, 500 non-fatal myocardial infarctions and 500 cases of stroke".

By 2030 [12], the European Union has set the goal of reducing its greenhouse gas emissions to at least 55% below 1990 levels. Measures have been adopted at the European level to combat maritime pollution. A cost-effective path has been foreseen to increase climate neutrality by 2050.

Directive 1999/32/CE is also of great importance for this purpose; it relates to the reduction in the sulfur content of certain liquid fuels. It has been substantially amended several times, most recently through Directive 2012/33/EU of 21 November 2012 on the content of sulfur of fuels for marine use, which entered into force on 17 December 2012.

Based on EC reports, the results deriving from the implementation of Directive 1999/32/EC since the last revision of 2012 (now codified as Directive (EU) 2016/802) are positive. There has been a significant reduction in $SO_2$ concentrations in coastal regions, particularly in the SECAs, while keeping the overall economic effects to a minimum (EC Report 2018).

Regulation (EU) 2016/1628 [13] concerns measures to be taken against the emission of gaseous and particulate pollutants produced by internal combustion engines intended for installation on non-road mobile machines.

Directive (EU) 2016/802 of the European Parliament and of the Council of 11 May 2016 concerns marine fuel consumption, sulfur content, implementation of cold ironing in European Ports, and setting of SECAs in the North and Baltic Seas, with a maximum of 0.1% sulfur content in the marine fuel consumed by vessels in these areas and areas outside SECAs (3.50%). In addition, starting from 1 January 2010, the maximum sulfur content limit in fuels should not exceed 0.1% for ships staying in European ports. The technology of cold ironing has been analyzed, and has been developing for about ten years, especially in North America and Europe.

The review carried out on the external costs paid particular attention to the description of the quantification of health costs through the use of the methodology described by CAFE [14], whose parameters have been adopted for the economic analysis carried out here.

## 4. Methodology

This research adopts an interdisciplinary approach based on scientific disciplines such as chemistry, environmental sciences, engineering, economics, and law. The methodology employs quantitative data adopting a bottom-up approach for the data collection on a local scale. For the quantification of health costs, reference was made to the "Clean Air for Europe" (CAFE) program, which defines a calculation method to estimate the external costs generated by some polluting agents such as NH3, NOx, $SO_2$, VOCs, and PM2.5. Their energy absorptions for the reference period were calculated through the information of ETAs and ETDs provided by the Genoa Port Authority and the type of ships. Consequently, the following was calculated:

- The emissions generated by the ships with the auxiliary engines turned;
- The electricity produced without alternative energy;
- The emissions deriving from the production of the alternative energy assumed in the energy mix.

Subsequently, assuming these different scenarios, an estimate was made of the emissions reduction derived from the use of the electrical connection on the quay.

Quantitative information concerning demand, duration of docking at the port, internal generation, and use of the external grid was obtained from primary and secondary sources. Primary information was collected using interviews with selected representatives of the Genoa Port Authority. Secondary information was gathered through an extensive desk search for journal articles from Science Direct, Google Scholar, and official organization websites such as IMO and DNV-GL.

The following data were calculated:

- The emissions generated by the ships with the auxiliary engines turned on;
- The emissions deriving from the production of electricity, either internal or external;
- The production of the alternative energy, assuming an energy mix that derives from a future consisting of 39 wind turbines located on different sections of the breakwater of the Port of Genoa for an installed power of 7.8 MW and an annual energy production forecast equal to about 12 GWh;
- Estimation of the reduction in emissions deriving from the electrical connection on the quay, assuming different scenarios;
- Assessment of the economic feasibility of the deployment of cold ironing.

## 5. Main Costs

Costs that were analyzed are listed below.

### 5.1. Transport Cost Externality

An external cost (or negative externality) is a cost that arises when a subject's social and economic activities have a negative impact on another person and when this impact is not fully justified or compensated by the first subject. This concept has been the subject of discussions by many experts since 1920 [15,16].

External costs have attracted particular attention internationally and at the Community level. In both areas, there has been full recognition of their importance and the considerable importance attributed to their evaluation.

As mentioned above, the emissions generated by maritime transport have a significant effect on air quality, especially in coastal areas [17,18]. Generally, ships use low-quality fuel to reduce costs, and this low quality is due to the presence of a high level of sulfur. Emissions of sulfur oxides (SOx) from transport account for around 60% of global SOx transport emissions. Emissions of nitrogen oxides (NOx) from transport account for about 15% of global anthropogenic NOx emissions and about 40% of global NOx emissions of goods transport. Shipping produces around 15% of the carbon dioxide ($CO_2$) emissions of goods transported globally (2–3% of total global $CO_2$ emissions). Regarding the sulfur

in the atmospheric particulate, the ships' emissions contribute 44% of its production [19]. Figure 2 shows the impact of shipping on the emissions of PM2.5 polluting agents.

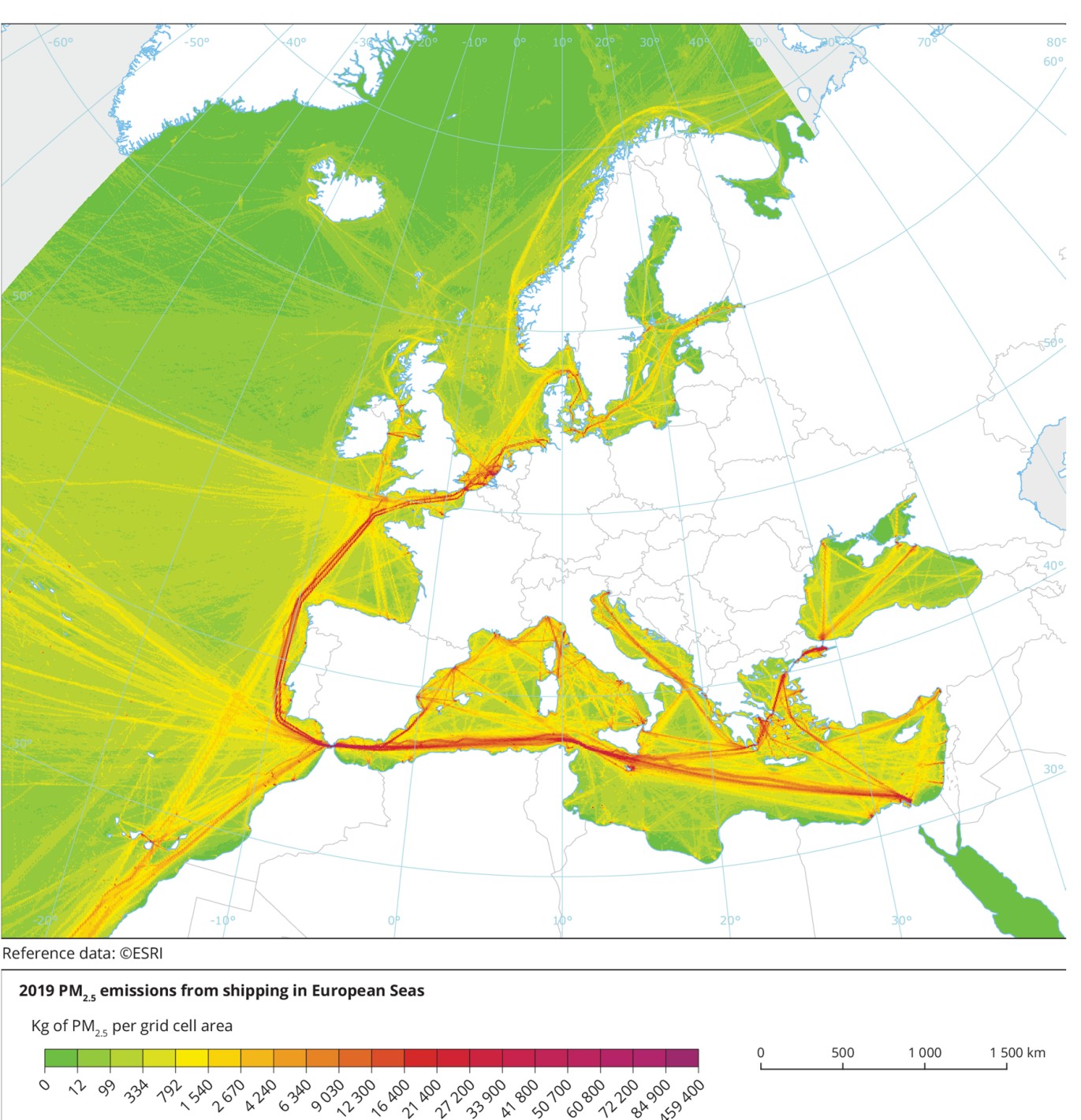

**Figure 2.** PM2.5 contribution from shipping in European seas in 2019. Source: EEA (European Environment Agency).

Emissions from engine combustion are divided into two broad categories:

- Emissions that directly influence the quality of the air and, therefore, with effects on a local scale (SOx, NOx, and PM);
- Emissions that directly influence global warming $CO_2$ and other greenhouse gases [20–23] with effects on a large scale.

*5.2. Health Costs and the CAFE Model*

Several studies carried out in recent years have provided solid evidence of the functional link between air pollution and adverse health effects for exposed populations [20–22].

A study by James J. Corbett shows that maritime transport causes about 60,000 deaths annually on a global scale, and that its concentration of effects is detected in the coastal regions facing the main trade routes.

The effects of higher mortality are found in Asia and in Europe, where the concentrations of PM2.5 are higher than those in other regions of the world.

The "Clean Air for Europe" (CAFE) program, which began in March 2001 and explored the problems linked to particulate matter and ozone, has focused on the development of a long-term and integrated thematic strategy that can counteract the damage caused by atmospheric pollution to the environment and human health.

As part of the development of the CAFE program, a significant contribution to the development of the abovementioned cost–benefit analysis was provided by a report under the program that provides a calculation method aimed at estimating the external costs generated by some polluting agents such as NH3, NOx, $SO_2$, VOCs, and PM2.5.

The analyses from this report are based on the methodology utilized within the external project funded by the EC DG Research. This methodology follows the steps and calculations listed below:

1. Pollutant emission;
2. Dispersion of pollutants;
3. The exposure of people, ecosystems, materials, etc.;
4. Quantification of impacts;
5. Evaluation of impacts.

## 6. Energy Consumption

*Evaluation of Energy Consumption*

A short overview of some characteristics of the container ship is provided because it represents the type of ship that most frequently visits the VTE. It is also shown how the conversion from GT to TEU was calculated, which is useful for applying energy absorption assumptions of the AP GE.

A survey [24] carried out by ABB on two container-ship groups (longer and less than 140 m), shows that the voltage in this type of vessel varies from 380 V to 6.6 kV, while most of the ships of higher tonnage use a voltage of 440 V. The voltage of 6.6 kV was detected only on ships built after 2001 for power services and primary distribution. The frequency is 50 Hz or 60 Hz.

Moreover, the container ships (with a length greater than 140 m) observed during the docking phase were characterized by an average demand of power equal to 2 MW and peak values equal to 8 MW. These values are high compared to the needs of the other types of ships covered by the same survey: for ro-ro ships, for example, a power requirement of 2 MW was estimated but with significantly lower peak values than the container vessel, i.e., 2 MW.

Precisely, in this case study, the adopted average energy absorption was that found from surveys carried out by the Port Authority (AP) of Genoa, which verified that the average energy absorption for each ship is as follows:

- Small–medium-size ships (<10,000 TEU): Energy absorption 900 kWh;
- Large-size ships (>10,000 TEU): Energy absorption 1500 kWh.

The data concerning the energy absorption of naval units were provided with reference to the TEU owned by each unit (Table 1). Regarding data relating to vessels that have visited the port, survey data expressed in gross registered tonnage (instead of TEU) were available, so it was necessary to identify the correlation relating to capacity (TEU/vessel) for GT, using a correlation coefficient R that is equal to 9.4 and showing a high value (close to 1; see Figure 3).

**Table 1.** TEU vs. GT. Source: Containerization International, 2009.

| TEU/Ship | Gross Tonnage (t) |
| --- | --- |
| 4035 | 50,657 |
| 4116 | 50,698 |
| 4196 | 45,000 |
| 4292 | 49,985 |
| 4296 | 49,985 |
| 4300 | 50,698 |
| 4306 | 50,698 |
| 4338 | 50,698 |
| 4437 | 52,181 |
| 5618 | 66,526 |
| 5618 | 66,590 |
| 6070 | 74,000 |
| 6070 | 79,702 |
| 6500 | 74,000 |
| 6500 | 74,642 |
| 6600 | 91,560 |
| 6600 | 93,496 |
| 6930 | 80,942 |
| 6978 | 80,654 |
| 8400 | 94,193 |
| 8400 | 98,400 |
| 8400 | 95,000 |
| 8600 | 91,427 |
| 8600 | 93,750 |
| 8600 | 106,700 |
| 9000 | 97,933 |
| 9000 | 99,500 |
| 9200 | 106,700 |
| 9200 | 95,000 |
| 12,508 | 156,907 |

Looking at the graph of Figure 3 the part above the line is a measure of the degree to which x and y vary together (using the deviations of each from their mean value). The part below the line is a measure of the degree to which x and y vary separately.

To further confirm the above data, it was considered useful to refer also to a study carried out by the Port Authority of Barcelona (Eco calculator 2013) [25] that analyzed 3296 available container ships. The ratio between GT and the number of TEUs was obtained by performing a statistical regression of the available data resulting from the following equation:

$$GT = -0.00016 \, TEU^2 + 13.284 \, TEU + 1696.27 \tag{1}$$

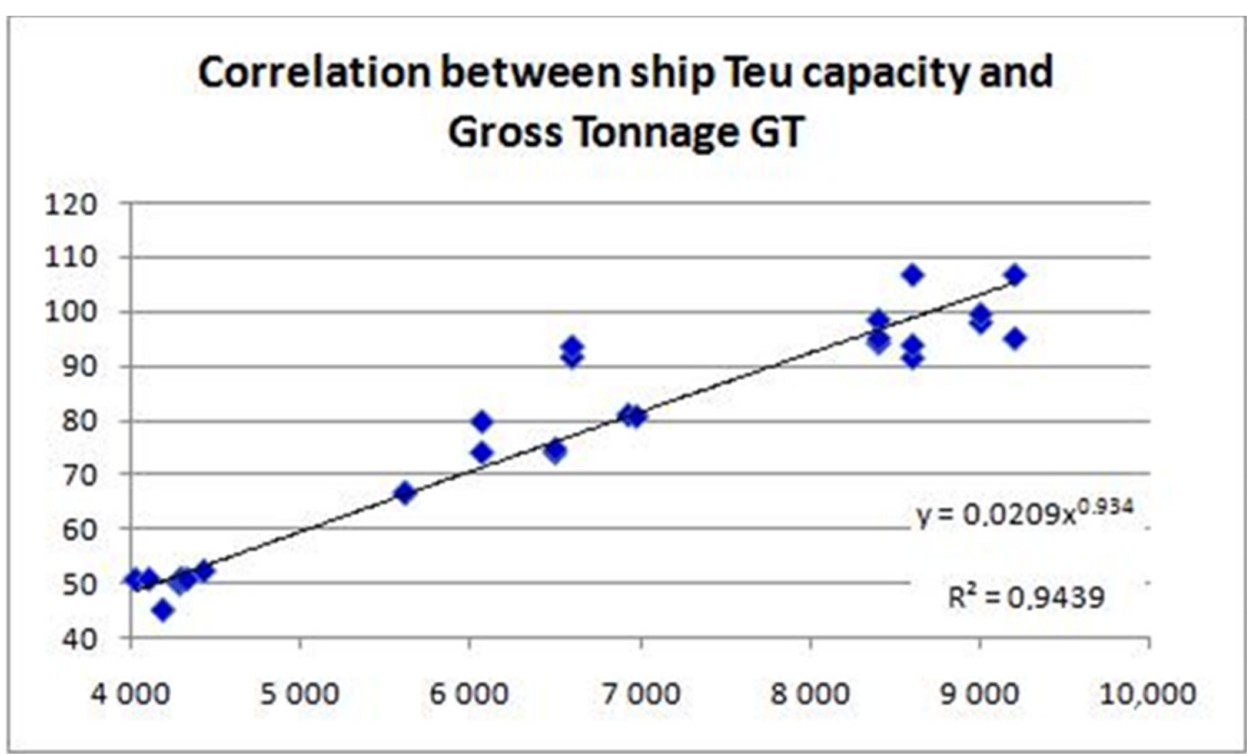

**Figure 3.** Correlation between TEU and GT. Source: Containerization International, 2009.

### 7. Case Study Port of Genoa

The port of Genoa is a seaport overlooking the Mediterranean Sea, and is the largest Italian port in terms of the volume of trade.

PSA Genova Pra' is part of the PSA INTERNATIONAL Group, a world leader in container terminal management. Active since 1992, it is the largest container terminal in the Port of Genoa and is a reference point for the Mediterranean area. Figure 4 below shows the calculation steps used for the cost–benefit analysis.

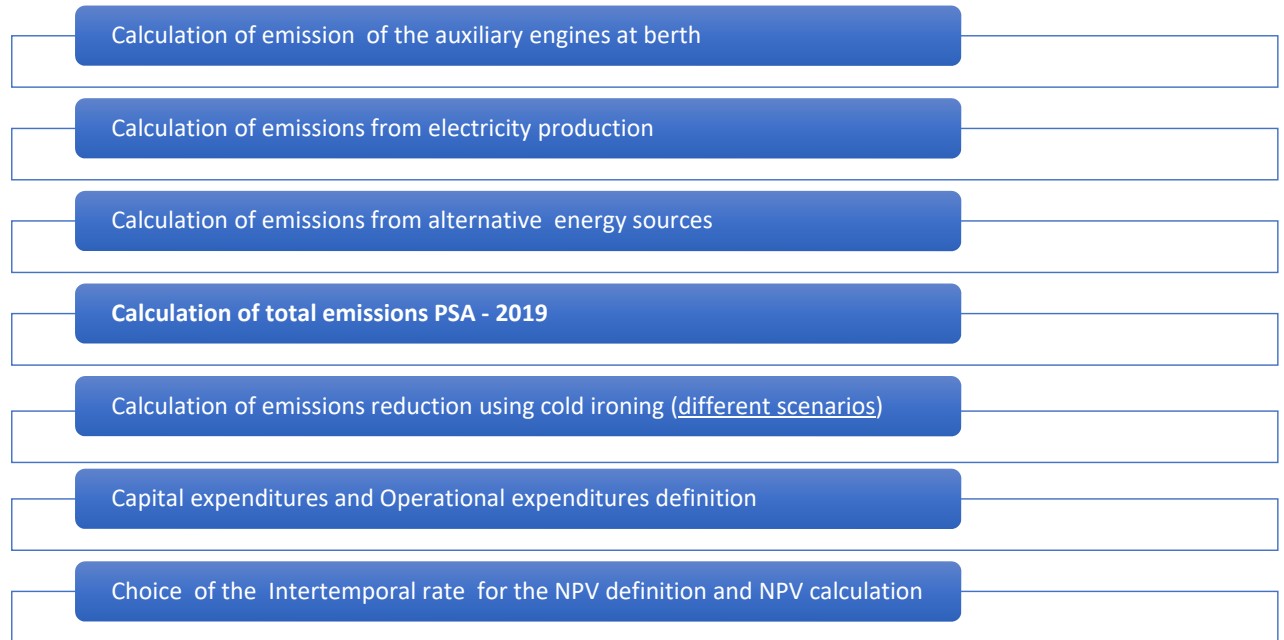

**Figure 4.** Case study methodology steps (source own elaboration).

*7.1. Total Energy Demand of Ships Hotelling at PSA−2019*

7.1.1. Input Data and Assumption

Energy Demand of Container ships Hotelling at berth—based on a study implemented by the port authority of Genoa:

- Small–medium-size ships (<10,000 TEU);
- Energy demand is 900 kWh;
- Large-size ships (>10,000 TEU);
- Energy demand is 1500 kWh;
- ETA and ETD;
- Hotelling hours are 17.991.

7.1.2. Output

The measured total and average energy demand of ships hotelled at PSA are:

Energy demand for the unit and for hotelling time/kw = Energy demand for unit kWh × Total hours of hotelling

- Total hours—17.991;
- Total energy demand (MWh)—20.231.

The calculation of the arithmetic average of the hours hotelled at the berth for the year 2019 was also performed:

$$\frac{1}{n}\sum_{i=1}^{n}x_i$$

The result is 29.74 h [arithmetic mean of number of hours and the number of docked ships].

*7.2. Calculation of Emissions in the Hotelling Phase Based on Engines Using 0.1% Sulphur*

7.2.1. Input

The input data used to calculate phased hoteling emissions for a ship with engines fueled with 0.1% sulfur fuel are shown in Table 2.

**Table 2.** Emissions in the hotelling phase based on factors from engines using 0.1% sulfur fuel.

| Nox (g/kWh) | $SO_2$ (g/kWh) | PM (g/kWh) | $CO_2$ (g/kWh) |
|---|---|---|---|
| 11.8 | 0.46 | 0.3 | 690 |

Source: Entec UK Ltd. (2005).

7.2.2. Output

Emissions per port call = (Hotelling time × single unit) × (Energy demand × single unit/kWh) × (Emission factors − auxiliary engines at berth, g/kWh)

The resulting average energy demand per call (MW/h) is 33.50 MW/h.

*7.3. Emissions Year 2019*

Table 3 shows the PSA total emissions in the hotelling phase, considering the use of auxiliary engines powered by fuel with 0.1% sulfur.

**Table 3.** PSA total emissions hotelling phase.

| Total PSA Emissions—Hotelling Phase (Using Fuels with 0.1% Sulfur) YEAR 2019 (Ton) | | | |
|---|---|---|---|
| $NO_x$ | $SO_2$ | PM | $CO_2$ |
| 238.73 | 9.31 | 6.07 | 13,959.51 |

Source: Own elaboration.

### 7.4. Calculation of Emissions Based on Factors Defined by the EU 25 Countries

The calculation was performed for electric energy production per average emission factors, and the source was ENTEC UK 2005) [26] for $CO_2$ (Source ISPRA2012) [27].

#### 7.4.1. Input Assumptions

Table 4 shows the assumptions used for the calculations.

**Table 4.** Average emission factors for electricity generation.

| Emission Factors for Electricity Generation—g/kWh of Electricity (Assumption) | | | |
|---|---|---|---|
| NO$_x$ (g/kWh) | SO$_2$ (g/kWh) | PM (g/kWh) | CO$_2$ (g/kWh) |
| 0.35 | 0.46 | 0.03 | 396.3 |

Source: Fonte: Entec UK Ltd. (2005).

#### 7.4.2. Output

Table 5 shows the emissions generated to generate the electricity.

**Table 5.** Total emissions for the production of electricity (TON)—Demand PSA 2019.

| NO$_x$ | SO$_2$ | PM | CO$_2$ |
|---|---|---|---|
| 7.08 | 9.31 | 0.61 | 8017.61 |

Source: Own elaboration.

### 7.5. Total Emissions Using Cold Ironing with Energy Production Based on the Average Emissions Factors (Source ENTEC)

#### 7.5.1. Input and Assumptions

Table 6 provides the emission factors when using shore power, compared to using fuels containing 0.1% sulfur content for ships' auxiliary engines.

**Table 6.** Emission factors with power supply from the shore power grid (compared to the use of fuels with 0.1% sulfur content for ship auxiliary engines).

| No$_x$ (g/kWh) | SO$_2$ (g/kWh) | PM (g/kWh) | CO$_2$ (g/kWh) |
|---|---|---|---|
| 0.39 | 0.46 | 0.03 | 395 |

#### 7.5.2. Output

Table 7 displays the total emissions of PSA during the hotelling phase, considering the utilization of cold ironing and electricity imported from the national grid based on average emission factors.

**Table 7.** PSA total emissions—hotelling phase using cold ironing and electricity import from the national grid as per average emission factors, EU 25 Countries (Entec 2005)—year 2019 (Ton).

| NO$_x$ | SO$_2$ | PM | CO$_2$ |
|---|---|---|---|
| 7.89 | 9.31 | 0.61 | 7.991 |

Source: Own elaboration.

### 7.6. Calculation of Emissions from Alternative Energy Sources Input and Assumption

The use of cold ironing and the import of energy from the national grid was considered (no internal generation).

#### 7.6.1. Input

Table 8 presents the emission factors of wind energy, taking into account its entire life cycle, including the emissions associated with its construction, operation, and de-commissioning.

**Table 8.** Emission factors of wind energy including its construction (g/kWh).

| Wind Energy Technology Lifecycle Emission Factors g/kWh | | |
|---|---|---|
| No$_x$ (g/kWh) | SO$_2$ (g/kWh) | CO$_2$ (g/kWh) |
| 0.02 | 0.06 | 9 |

Source: Own elaboration.

### 7.6.2. Output

Data necessary for calculations of emissions as a consequence of this technology were found in a Renewable Energy Foundation study. CUT emissions achieved by this solution were calculated.

Table 9 displays the emission factors for wind power plant production equivalent to 12,000 MWh, providing insight into the factors associated with emissions during renewable energy generation.

**Table 9.** Factors of wind power plant emissions for production equal to 12,000 MWh (t).

| | NO$_x$ (g/kWh) | SO$_2$ (g/kWh) | CO$_2$ (g/kWh) |
|---|---|---|---|
| **Wind power plant emission factors for production equal to 12,000 MW (t)** | 0.24 | 0.72 | 108 |

Source: Own elaboration.

### 7.7. Calculation of Cold Ironing Emission Factors Using Different Energy Sources

A potential project exists relating to a wind park turbine located on different sections of the breakwater of the Port of Genoa for an installed power of 7.8 MW and an annual energy production forecast equal to about 12 GWh.

Therefore, the level of emissions using this wind park was considered.

### Output

Table 10 showcases the emissions associated with cold ironing, considering various energy sources utilized.

**Table 10.** Cold ironing emissions with different energy sources.

| Cold Ironing Emission Factors with Different Energy Sources | NO$_x$ (g/kWh) | SO$_2$ (g/kWh) | PM (g/kWh) | CO$_2$ (g/kWh) |
|---|---|---|---|---|
| **Electricity production Eu 25 entec g/kWh** | 0.39 | 0.46 | 0.03 | 395 |
| **Wind emission factors g/kWh** | 0.02 | 0.06 | n.d. | 9 |

Source: Own elaboration.

### 7.8. Cut Emissions Scenarios

This section presents different cut emissions scenarios based on the data calculated in the abovementioned scenarios.

Scenario one illustrates the reduction in emissions using cold ironing (100% ships) with electricity production based on the average emissions factors used for EU countries.

Scenario two analyzes the reduction and considers the production of the future wind farm. The basis of wind energy is the harnessing of the wind's power through wind turbines. It is one of the cleanest forms of energy.

In both cases, the ship emissions reduction is remarkable, with more values adopting an energy mix, including wind power plants. The main disadvantage is that this generation may be intermittent, so import from an external grid or other forms of generation and storage are necessary and must be managed in a cost-effective way.

Table 11 presents two scenarios illustrating the potential emission reductions achieved through the implementation of cold ironing in the PSA for 2019. In scenario 1, assuming 100% of ships utilize cold ironing based on electricity production with average emission factors for EU 25 countries, a significant reduction in emissions is observed. However, in scenario 2, where the hypothetical energy mix is considered, the benefits in terms of emission reductions are even more significant. Scenario 2 indicates a substantial 52% variation in $SO_2$ emissions, while Scenario 1 shows no variation for the same emission factor.

**Table 11.** Scenario 1, Scenario 2.

| SCENARIO 1—100% SHIPS USE COLD IRONING FOR THE YEAR 2019 Electricity production as per average emission factors for EU countries 25 (Entec 2005) | | | | | |
|---|---|---|---|---|---|
| | Energy Demand (MW) | $NO_x$ (t) | $SO_2$ (t) | PM (t) | $CO_2$ (t) |
| Ships Emissions with running auxiliary engines | 20,231 | 239 | 9 | 6 | 13,960 |
| Ship Emissions using cold ironing | 20,231 | 8 | 9 | 1 | 7991 |
| DIFFERENCE | | 231 | - | 5 | 5968 |
| **% VARIATION** | | **97%** | **0%** | **90%** | **43%** |
| SCENARIO 2—100% SHIPS USE COLD IRONING FOR THE YEAR 2019 production of electricity as per the hypothesized energy mix | | | | | |
| | Energy Demand (MW) | $NO_x$ (t) | | $SO_2$ (t) | $CO_2$ (t) |
| Ships Emissions with running auxiliary engines | 20,231 | 239 | | 9 | 13,960 |
| Ship Emissions using cold ironing with hypothesized energy mix | 20,231 | 3.44 | | 4.49 | 3347 |
| DIFFERENCE | | 235 | | 5 | 10,613 |
| **% VARIATION** | | **99%** | | **52%** | **76%** |

Source: Own elaboration.

### 7.9. Health Cost Scenarios

As illustrated before, the CAFE analysis was used to determine health cost savings using cold ironing.

Table 12 shows different scenarios of using cold ironing, indicating the saving in terms of health costs.

**Table 12.** Different scenarios, 2019.

| MARGINAL DAMAGES POLLUTING AGENT FOR TON OF EMISSION | |
|---|---|
| $NO_x$ | € 8600 |
| $SO_2$ | € 9300 |
| $PM_2$ | € 52,000 |

**Table 12.** *Cont.*

| SCENARIO 1—100% SHIPS USE COLD IRONING FOR THE YEAR 2019 electricity production as per average emission factors for EU countries 25 (Entec 2005) | | | |
|---|---|---|---|
| | Nox (t) | SO$_2$ (t) | PM (t) |
| Health costs deriving from emissions from ships with auxiliary engines running | € 2,053,060 | € 86,549 | € 315,606 |
| Health costs deriving from Emissions Ships using cold ironing | € 67,855 | € 86,549 | € 31,561 |
| DIFFERENCE | € 1,985,204 | €- | € 284,046 |
| **% VARIATION** | **97%** | **0%** | **90%** |
| Total health cost savings by using cold ironing | | | **€ 2,269,249.98** |
| **SCENARIO 2—Health costs in case 60% of ships use cold ironing (with assumed energy mix)** | | | |
| | Nox (t) | SO$_2$ (t) | PM (t) |
| **All ships generate energy by means of auxiliary engines (0.1% sulfur)** | | | |
| Emissions from ships with running auxiliary engines | € 2,053,060 | € 86,549 | € 315,606 |
| **60% of the ships adopt cold ironing (production of electricity as per the hypothesized energy mix) The remaining ships generate energy through the auxiliary engines (0.1% sulfur)** | | | |
| | Nox (t) | SO$_2$ (t) | PM (t) |
| Emissions from ships with running auxiliary engines | € 821,224 | € 34,620 | € 126,243 |
| Emissions Ships using cold ironing | € 40,713 | € 51,929 | n.d. |
| TOTAL | € 861,937 | € 86,549 | |
| DIFFERENCE | € 1,191,123 | € - | n.d. |
| **% VARIATION** | **58.02%** | **0%** | **n.d.** |
| Total health cost savings by using cold ironing | | | **€ 1,191,122.57** |

Source: Own elaboration.

## 8. Conclusions

The Onshore Power Supply system for ships is expected to be an environmentally effective solution that cooperates significantly to reduce or virtually eliminate air pollution in ports and related areas, and noise deriving from the auxiliary engines.

Cold ironing or the OPS is basically an anti-pollution measure that is able to reduce air pollution produced by ship auxiliary generators through the use of shore electric power as an alternative. The ship's power load is transferred to the shoreside power supply without an impact on onboard services.

The adoption of the Onshore Power Supply to reduce polluting emissions is desirable. It is also important to consider that the benefits resulting from this analysis are still underestimated since, due to the lack of authoritative data relating to $CO_2$ emissions (data not calculated within the CAFE methodology), it was not possible to quantify their resulting health costs.

To encourage this development, it is necessary for all the actors involved to have greater sensitivity towards environmental problems, and to force politicians to follow this approach. This could be a solid starting point.

In the design phase, it is advisable to evaluate the best mix (in terms of emission reductions, reliability, and resilience) of conventional electricity production with alternative energy sources to achieve the goal of maximizing the reduction in emissions.

In the future, the synergy between political decision makers, ports, and shipowners will be necessary to reach a solution compatible with the environment and sustainable for the parties involved.

**Author Contributions:** Conceptualization, M.C.; methodology, M.C.; software, M.C.; validation, D.D.; formal analysis, M.C.; investigation, M.C.; resources, G.F.; data curation, D.S.; writing—original draft preparation, M.C.; writing—review and editing, G.F.; visualization, D.D.; supervision, F.B. All authors have read and agreed to the published version of the manuscript.

**Funding:** This research received no external funding.

**Data Availability Statement:** The data that support the findings of this study are available from the corresponding author, M.C., upon reasonable request.

**Conflicts of Interest:** The authors declare no conflict of interest.

## Glossary

| Term | Description |
| --- | --- |
| AP | Port Authority |
| CAFE | Clean Air for Europe |
| DNV HG | Horizon Graphic and Det Norske Veritas |
| INFO | European Alternative Fuels Observatory |
| EMTER | European Maritime Transport Environmental Report |
| ENTEC | Environmental and Engineering Consultancy |
| ETA | Expected time of Arrival |
| ETD | Expected time of departure |
| GE | Genoa |
| GHG | Green House gas |
| GT | Gross tonnage |
| IMO | International Maritime Organization |
| MARPOL | International Convention for the Prevention of Pollution |
| NO2 | Nitrogen oxides |
| OPS | Onshore power supply |
| PM | Particulate matter |
| SECA | Sulfur emission control areas |
| SOX | Sulfur dioxide |
| TEU | Twenty-Foot Equivalent Unit |
| VTE | Voltri Terminal Europe |

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
