# Peer review of "Cold Ironing: Socio-Economic Analysis in the Port of Genoa"

_logistics, 2023_

Round 1

Reviewer 1 Report

The paper is properly organized. A decent methodology design is used. Meanwhile, some figures and tables are blurred, e.g., Figure 1 and Figure 5. Please impove the resolutions these figures. 

Author Response

Thanks for your comments. Low resolution tables and figures have been replaced. MInor English error has beencorrected

Reviewer 2 Report

1) The authors should improvise the introduction section. The flowability is pretty poor. 

2) A literature review section needs to be added to reveal clearly the research gaps and hence the novelty of the presented work. 
3) The objectives of the presented work are not clear. 

4) A flow chart for the methodology will improvise the comprehensibility of the work. 

5) The calculations are vague and should be improvised. 

6) Figures are not clear. Request to provide high resolution images.

Author Response

Thanks for your comments. Introduction has been partially rewritten and objectives narrowed to be adherent
with the text. High resolution figures and tables have been included. Calculation details of CAFE- methodology
have not been included to avoid excessive length, however they are available to interested readers

Reviewer 3 Report

In my opinion, the authors did not want to publish a scientific paper. The manuscript seems to represent the summary of a project. Mathematical methods and models are not presented or simple relationships are used that do not reflect the quality of such research. I have a request for the authors to reorganize their work, possibly to co-opt in the team an author with experience in writing scientific papers.

Author Response

Thanks for your comments. The paper does not want to develop novelty models, but makes use of an
established methodology to calculate the effect of OPS in different scenarios of VTE terminal near Genoa, using
different scenarios based n the use of traditional generation or renewables as hat are forecasted in an new
project. Its target is to show to practitioners the quantitative advantages in a few defines scenarios Calculation
details in excel format are available on request of interested readers

Reviewer 4 Report

This paper is interesting. Somehow it did not connect relevant artifacts or variables while writing and explaining the paper. Images are blurry and low resolution and did not explain well. 

As a reviewer, I did not check the plagiarism. It is requested to do it before going for final publication. This paper lack of communication to the relevant and general readers. It needs to explain with necessary and meaningful words to understand contribution. 

Table 11 and 12 did not explain well. 

Author Response

Thanks for your comments. High resolution images and tables have been included when necessary. A few
sections including introduction have modified to improve understandability for general readers. A glossary has
been included to improve readability

Round 2

Reviewer 2 Report

Can be accepted in present form

Author Response

Many thanks for your review. 

Based on this we have improved the paper.

Regards.

Monica Canepa

Reviewer 3 Report

Authors have made a series of improvements following my recommendations. I hope that the following disseminations will be carried out in a more professional manner.

Author Response

(The authors gave the same response as above.)
